# Counting on Numbers—Numerical Abilities in Grey Bamboo Sharks and Ocellate River Stingrays

**DOI:** 10.3390/ani11092634

**Published:** 2021-09-08

**Authors:** Nils Kreuter, Nele Christofzik, Carolin Niederbremer, Janik Bollé, Vera Schluessel

**Affiliations:** Institute of Zoology, University of Bonn, Poppelsdorfer Schloss, Meckenheimerallee 169, 53115 Bonn, Germany; nkreuter93@gmail.com (N.K.); nele.christofzik@gmx.de (N.C.); caro.97@web.de (C.N.); janik.bolle@gmail.com (J.B.)

**Keywords:** cognition, visual discrimination, elasmobranch, quantity discrimination

## Abstract

**Simple Summary:**

This study examined the quantitative discrimination abilities (using two-dimensional objects) in a shark and a stingray species. Both species underwent a training procedure, followed by a series of transfer tests designed to investigate whether they could extrapolate and apply learned knowledge to a set of new quantity discrimination tasks. Sharks and rays successfully mastered the training tasks as well as most of the transfer tests. This included the discrimination of 4:1, 5:2 and 7:5. The present study is the first to describe numerical abilities in elasmobranchs, in which any potentially confounding, non-numerical factors (e.g., size and area) were controlled for. This study adds to the growing number of studies on fish that are key to understand the evolution and development of cognition in vertebrates.

**Abstract:**

Over the last decade, studies examining the cognitive abilities of fish have increased, using a broad range of approaches. One of the foci has been to test the ability of fish to discriminate quantities of items and to determine whether fish can solve tasks solely on the basis of numerical information. This study is the first to investigate this ability in two elasmobranch species. All animals were trained in two-alternative forced-choice visual experiments and then examined in transfer tests, to determine if previously gained knowledge could be applied to new tasks. Results show that the grey bamboo shark (*Chiloscyllium griseum*) and the ocellate river stingray (*Potamotrygon motoro*) can discriminate quantities based on numerical information alone, while continuous variables were controlled for. Furthermore, the data indicates that similar magnitudes and limits for quantity discrimination exist as in other animals. However, the high degree of intraspecific variation that was observed as well as the low rate of animals proving to be successful suggest that the ability to discriminate quantities may not be as important to these species as to some other vertebrate and invertebrate species tested so far.

## 1. Introduction

The ability to discriminate quantities has shown to be a widespread cognitive skill in a variety of animals including mammals [1], insects [2], birds [3], fish [4], amphibians [5] and reptiles [6]. This is not surprising, as quantity discrimination is advantageous for many behaviours including foraging [7,8], mating [9,10] and predator avoidance [11,12]. There is an ongoing debate in the literature on how quantities are perceived and processed [13]. On the one hand studies suggest that two core systems exist, which play a prominent role in the ability to quantify larger and smaller numerical ratios—the object file system (OFS) and the approximate number system (ANS) [14,15,16,17]. The OFS is faster, more accurate and enables identification of individual objects with a one item difference being enough for distinction. However, it is limited to a small number (≤5) of presented objects [15,18,19,20]. The ANS is associated with estimating larger numerical magnitudes (>5) based on Weber’s Law ‘of the just noticeable difference’ [15,21,22]. On the other hand it has been suggested that these two are in fact not individual systems but that the ANS is in charge of processing both small and large quantities [23,24]. Studies in mammals, fish and other animals have provided evidence to support either of the two hypothesized systems [4,25,26,27]. Agrillo et al.’s review argues that the results of most studies on fish are supporting the two-system hypothesis [13]. Therefore, the present study is focusing its analysis based on the hypothesized two core system and will not engage in debating whether a two core or single core system exists.

In the past, the research focus concerning numerical abilities was mainly on primates and birds; however, an increasing number of studies on a few fish species emerged over the last decade [28]. The majority of these studies involved guppies (*Poecilia reticulata*), angelfish (*Pterophyllum scalare*) and mosquitofish (*Gambusia holbrooki*) [28]. Guppies have shown to successfully discriminate 5 vs. 4 items while three-spined sticklebacks even distinguished 7 vs. 6 [12,29]. For larger quantities (>5) fish, like other vertebrates, seem to rely on Weber’s law and are only able to distinguish between the size of two sets of given objects when the ratios are lower. Mosquitofish and guppies successfully discriminated quantities up to 12 vs. 8 (mosquitofish) and 12 vs. 6 (guppies), while both species failed at 12 vs. 9 [30,31]. Ratios tend to have an important influence on the species’ ability to discriminate quantities, especially when the numbers fall within the ANS. Mosquitofish showed a decrease in performance when the ratio increased and the numerical distance between two sets of stimuli decreased [30]. Animals have been shown to use both numerical (discrete) and continuous variables (e.g., surface area, size, density etc.) to solve quantitative tasks [32]. Discrete quantities are countable and represented by a certain value while continuous variables are a measurement of a specific quantity that correlates with the number of items that are presented. This means for example that the surface area of a fixed object increases with its numerosity and therefore needs to be controlled when investigating the ability to discriminate discrete quantities. This control prevents the influence of actively avoiding or choosing the larger area occupied by the larger amount of stimuli presented [32]. Agrillo et al., showed that the availability of both continuous and numerical information combined leads to a more accurate outcome than one of them alone [33]. Accordingly, when testing numerical discrimination, the use of continuous variables needs to be controlled for, and this “represents one of the most critical issues in this research field” [34]. In one of the first studies on fish, where continuous quantities were controlled for, mosquitofish demonstrated the ability to discriminate 2 vs. 3 [35]. Since then, discriminating abilities relying only on numerical information have been demonstrated in guppies (*Poecilia reticulata)* [36,37], angelfish (*Pterophyllum scalare*) [38], Siamese fighting fish (*Betta splendens*), zebrafish (*Danio rerio*), redtail splitfins (*Xenotoca eiseni*) [39], goldfish (*Carassius auratus*) [40] and a blind cavefish (*Phreatichthys andruzzii*) [41].

Several elasmobranch species, including the grey bamboo shark (*Chiloscyllium griseum*) and the ocellate river stingray (*Potamotrygon motoro*), have been part of a range of cognition experiments and have shown various visual discrimination as well as orientation abilities (see reviews [42,43,44,45,46,47,48,49,50,51,52,53]). Adding to that, freshwater stingrays (*Potamotrygon castexi*) are able to use water as a tool to extract food from a tube [54]. Even though the class of Chondrichthyans is at least 450 million years old and represents the oldest lineage of vertebrates [55], only one study demonstrating the ability to discriminate quantities in Port Jackson sharks (*Heterodontus portusjacksoni*) has been published [56]. The overall aim of the study was to investigate potential influences of elevated ocean temperature on cognitive abilities. During the research, continuous variables were not controlled for and therefore only the low cognitive load to distinguish quantities was shown.

For the current study, *C. griseum* and *P. motoro* were trained to visually discriminate quantities, as both species have shown to be well suited for visual cognition experiments. By training animals, opposed to testing for spontaneous choice, one can control all variables potentially influencing the animal’s choice [34]. In previous studies, grey bamboo sharks were able to distinguish between symmetrical and non-symmetrical shapes, various stationary objects (including geometric forms) as well as moving stimuli and categorized two-dimensional objects [49,52,57]. They also performed well in a range of optical illusion experiments [50,58]. In recent studies a wide range of visual discrimination abilities were shown in *P. motoro*, including first evidence for memory retention [45], serial reversal learning [46] and colour discrimination [45]. In addition, *C. griseum* and *P. motoro* both seem to be able to use spatial maps to orientate, can use different orientation strategies and memorize spatial tasks over a period of several weeks or months [51,59].

Individuals of both species underwent similar training procedures for this work and were subsequently tested on whether they could transfer numerical abilities they have previously learned to unknown ratios. This study is the first to examine the ability of elasmobranchs to use numerical information in which continuous variables were controlled for. It adds valuable insights to the field of shark and ray cognition and to understanding the evolution and development of cognition in vertebrates.

## 2. Materials and Methods

### 2.1. Housing and Experimental Facilities

The sharks were kept in five interconnected aquaria (each 300 L) and two 450 L aquaria, either individually or as pairs. All tanks were connected to an additional tank equipped with a filter unit, a protein skimmer and a UV lamp. To ensure constant water parameters throughout the setup (ca. 1.0217 kg salt/dm^3^; conductance: about 50 μS/m; 25 ± 1 °C; pH 8, KH 9–12°, NO_2_^−^ < 0.2 mg/L), parameters were checked twice a week, adjusted using Aqua Medic AB Reef Salt and Reef Life System Coral B buffer (AB Aqua Medic GmbH, Bissendorf, Germany). Weekly water changes (about 20% of total volume) were conducted. Experiments took place in a separate tank (50 cm ∗ 50 cm ∗ 100 cm) (Figure 1a), filled with water from the housing-system every morning and emptied and rinsed at the end of each day. The experimental tank was divided into a Starting Compartment (SC) and a Decision Compartment (DC). Both were separated by a grey PVC panel containing a hand-operated guillotine door (Figure 1a). All walls and the bottom side were covered with blue non-transparent adhesive foil on the outside of the tank to prevent the sharks from any visual disturbance. The lower half of the wall at the end of the DC was left uncovered to allow light from a projector (Optoma ES521, Optoma Corp., Taipei, Taiwan) to hit a milky colored plastic pane (50 cm ∗ 21.5 cm) on which the two-dimensional stimuli were displayed. The stimuli were prepared and presented as a PowerPoint presentation (ver. 16.24) and operated from a laptop next to the experimental setup. Both stimuli were arranged on the same slide during each trial. To ensure clear lateral decision-making, the milky pane featured a separating 10 cm broad and 9 cm high divider in its center. The divider’s breadth also indicated the beginning of the Decision Area (DA) at which a choice was counted if the animal passed over it. To distinguish the DA from the rest, the tank’s underside in the DA was covered in white foil. This created a line to ensure unambiguous data collection of the decision-making process. A webcam (Logitech c170, Logitech, Lausanne, Switzerland) was mounted on the ceiling above the setup to allow for visual observations of the shark during the experiment. It was only used to observe the shark’s choice, without the shark being able to see the experimenter during the trial.

Each stingray group (5 and 9 individuals) was kept in a large wooden tank (233 cm ∗ 233 cm ∗ 45 cm) lined with black pond foil covered in sand to enable rays burying themselves (Figure 1b). EHEIM Jäger Aquarium Heaters, two airstones, a pump and filter system and a weekly water change ensured steady water quality (conductance: 380–420 μS/m; 28 ± 1 °C; pH 6–8, KH 2–7°, NO_2_^−^ < 0.3 mg/L). Parameters were checked twice a week and adjusted using Aqua Medic AB Reef Salt and Reef Life System Coral B buffer (Aqua Medic, Germany). Experiments took place in a separate compartment (71 cm ∗ 125 cm ∗ 28 cm) within the pool (Figure 1b). Its partitioning was the same as the one used for the sharks, with a starting box for the rays added to the SC. The set-up only really differed in the way the stimuli were presented. Instead of using a projector to display stimuli, the experimenter changed a set of printed and laminated cards by hand (due to the wooden nature of the tank).

### 2.2. Experimental Procedure

All animals were trained in two-alternative forced-choice visual experiments (for an overview of all experiments see Table 1 in the Section 3). Each experiment consisted of up to 30 sessions with ten trials per session, in which a defined learning criterion (LC) needed to be met. To reach the LC, the animals had to make at least seven out of ten correct choices (70%) in three consecutive sessions before reaching 30 sessions (c ^2^ (1) ≤ 0.05). This predefined criterion ensured statistical significance in choice when reaching the LC. For all animals that completed an experiment, a two-tailed binomial test (95% confidence interval) was run in R (ver. 1.4.1103) to assess whether the sharks had a significant choice towards either the negative (lower quantity) or positive (higher quantitiy) stimulus.

An animal was placed in the SC of the respective experimental tank and given a few minutes to acclimatize. A trial commenced once the shark or ray crossed the opening of the guillotine door with the tip of its head, entering the experimental area. A choice was made by crossing the DA line with the tip of the head on the respective side (Figure 1). The time was stopped manually with a stopwatch (CG-501, Genutek Electronics Co. Ltd., Guangdong, China). A maximum time of 120 s was allowed for the decision-making process. When choosing correctly, the animal was fed and given a couple of seconds in front of the stimulus. A wrong choice was not rewarded with food and the animal was immediately guided back to the SC using a rubber kitchen scraper. To ensure olfactory cues were not influencing the experiment, the water was stirred after each trial. An inter-trial time of 30 s, which started when the shark or ray had returned to the SC and the door was shut, separated the trials. After the experiment, the animals were transferred or guided back to their resident tank. Sharks and rays underwent up to two experimental sessions per day, six times a week. The time between the two daily sessions for one individual was at least five hours.

### 2.3. Experimental Animals

For this study, two juvenile grey bamboo shark (*Chiloscyllium griseum*) groups and two ocellate river stingray (*Potamotrygon motoro*) groups were trained in an experimental procedure. Animals were divided into separate groups due to housing limitations and the experiments taking place over the period of two years on different individuals. Except for one group of rays (Group 3), all individuals were experimentally naïve and distinguishable by phenotypic characteristics (for details on size and age see Section 2.3.1 and Section 2.3.2). Food consisted of pieces of shrimp, fish, squid and earthworms and was provided during experiments only. Once a week, ten drops of a vitamin solution (Atvitol, JBL GmbH and Co. KG, Neuhofen, Germany) was added to the food. By using automatic time switches, all animals were kept in a 12 h light and 12 h dark cycle.

#### 2.3.1. Grey Bamboo Shark (*Chiloscyllium griseum*)


*(Group 1): 8 individuals (7 females, 1 male), head-tail length 35–40 cm, 1 session per day*


The first group of sharks was trained to discriminate 4 vs. 1 geometric symbols (square, circle, triangle) where all continuous variables (color, surface area, density (space occupied on slide) and geometry) were controlled for (except size of symbols) and choosing the higher quantity was rewarded with food. The first transfer test was therefore controlled for size and all stimuli had the same individual size (Figure 2e). Twenty transfer test trials for each shark were implemented. For the second part of the experiment the sharks were trained to discriminate 5 vs. 2 geometric symbols where all continuous variables (size, color, surface area, density (space occupied on slide) and geometry) were controlled for and choosing the higher quantity was rewarded with food. Transfer tests for this experiment consisted of unifying the stimuli color, showing non-geometric stimuli, reducing the surface area of the positive stimulus and examining ratios of 0.57 (7 vs. 4), 0.6 (5 vs. 3) and 0.8 (5 vs. 4) (Figure 2).


*(Group 2): 7 individuals (5 females, 2 males), head-tail length 25–35 cm, 2 sessions per day*


The second group of sharks underwent training to discriminate 4 vs. 1 and 5 vs. 2 geometric symbols with the color not being controlled for. As none of the sharks was able to reach the LC the experiments were further simplified following Agrillo et al.’s and Bisazza et al.’s approaches [33,36]. A second 5 vs. 2 and a 2 vs. 1 training experiment was implemented with the surface area and the color not being controlled for. A repetition of the Pretraining was conducted after finishing all regular experiments to exclude any visual or cognitive impairments throughout the study.

#### 2.3.2. Ocellate River Stingray (*Potamotrygon Motoro*)


*(Group 3): 5 individuals (1 female, 4 males), disk diameter 17–22 cm, 2 sessions per day*


The first group of stingrays consisted of animals that had previously been trained on a different non-numerical experiment, including the Pretraining procedure. For this study they underwent training to discriminate 5 vs. 2, 3 vs. 6, 4 vs. 7, 5 vs. 7, 6 vs. 7 and 7 vs. 2 while only the surface area was not controlled for. In the subsequent transfer tests the surface area was controlled for and the rays were tested whether they could still discriminate the quantities they were trained on.


*(Group 4) 9 individuals (5 females, 4 males), disk diameter 11—14 cm, 2 sessions per day*


The second group of stingrays were only trained on 4 vs. 1, 5 vs. 2 while continuous variables (size, colour, surface area, density (space occupied on slide) and geometry) were controlled for. During the transfer tests, a variety of different ratios were tested to examine the ability of the rays to discriminate quantities based solely on numerical information. In addition to the tested ratios, a transfer test was designed to quantify whether the animals used a relative or absolute approach to discriminate quantities.

### 2.4. Experiments

#### 2.4.1. Pretraining

All animals were tested in a “Pretraining” experiment to ensure their cognitive and visual wellbeing and their habituation of the experimental procedure. Animals needed to differentiate between a black circle and a blank, by choosing the side on which the black circle was displayed (Figure 2a). The correct choice was rewarded with food. All animals were examined during the Pretraining and evaluated whether they successfully managed to reach the LC before the 30th session. Those individuals that were experimentally naïve and did not reach the LC before the end of the 30th session, were given an additional 15 sessions after which their performance was analyzed. If a statistically significant choice (*p* < 0.05) for the correct stimulus occurred, they were permitted to the subsequent training sessions, else they were excluded from all experimental procedures (Figure 3).

#### 2.4.2. Numerical Discrimination Training

In the numerical discrimination training, two different stimuli that differed in the number of objects displayed were presented to the animals. Four different randomized object/side orders were developed and circulated to prevent animals from simply memorizing the order of the presented stimuli. During a session, each stimulus was presented five times on the right and five times on the left side, but never more than twice consecutively on the same side. For each group, a different set of symbols (but all consisting of geometric forms) was used. Once the LC was met and animals performed consistently above chance level, an 80% rewarding scheme was implemented, i.e., animals were maximally rewarded in 8 out of 10 correct trials. Two trials were randomly assigned to not be rewarded before each session, irrespective of actual choice. This procedure prepared animals for transfer testing, in which trials were generally unrewarded, by preventing a strong association of ‘no food reward’ with an ‘incorrect’ choice (which could have kept animals over time from participating in these trials).

#### 2.4.3. Transfer Tests

Once the LC was met, and the 80% rewarding scheme implemented, transfer testing began. Transfer tests were designed to examine the animal’s ability to transfer and extrapolate what it learned during training onto a new task. The regular training trials continued, but one or two transfer tests were added to the regular ten-trial session and were not rewarded. Transfer tests were carried out to examine (i) the ability to discriminate low and high ratios, (ii) a possible influence by shading and size, (iii) the ability to discriminate quantities of non-geometrical stimuli and (iv) whether a relative or absolute strategy was used.

## 3. Results

### 3.1. Pretraining

For group 1 six out of eight sharks reached the LC within 27 ± 7 sessions and six out of seven sharks from group 2 within 12 ± 9 sessions. All five rays of group 3 obtained the LC in 7 ± 10 sessions and eight out of nine rays from group 4 in 28 ± 17 sessions. It needs to be considered that the rays of group 3 were not experimentally naïve and have been previously trained in this setting. An exemplary graph of a Pretraining learning curve in a grey bamboo shark is shown in (Figure 4), where the LC was reached in the 13th session.

### 3.2. Sharks

#### 3.2.1. Group 1

Seven out of eight sharks were successfully trained to discriminate 4 vs. 1 within 30 sessions. On average the sharks needed 14.57 ± 7.56 sessions to reach the LC. A graph showing the performance of an individual shark during training and subsequent training while transfer tests were added can be found in Figure 5. The seven sharks were permitted to take part in twenty transfer tests each, where the area of the single negative stimulus was adjusted to the same size as one of the positive ones. Pooled as a group, the sharks chose the higher numerosity significantly more often (*n* = 7, *p* < 0.0001) (Figure 6). Three out of eight sharks were successfully trained to discriminate 5 vs. 2 within 30 sessions. On average, the sharks needed 17.33 ± 10.01 sessions to reach the LC. Subsequently, all three individuals each took part in twenty trials for each of the following transfer tests. Pooled as a group, the sharks chose the higher numerosity significantly more often while the stimulus color was the same (*n* = 3, *p* < 0.0001), non-geometric stimuli were shown (*n* = 3, *p* < 0.0001), the positive stimulus was reduced in size (*n* = 3, *p* < 0.0001) and the ratio was increased to 0.6 (5 vs. 3) (*n* = 3, *p* < 0.0001) and 0.57 (7 vs. 4) (*n* = 3, *p* < 0.001). No significant choice for the higher numerosity was shown for 5 vs. 4 (*n* = 3, *p* = 0.155) (Figure 6).

#### 3.2.2. Group 2

Six out of seven sharks passed the Pretraining test but were not able to be successfully trained on specific ratios, even after they were provided with additional continuous variables to ease the ability to discriminate quantities (following [33]). A repetition of the Pretraining was conducted after finishing all regular experiments to exclude any health-related, visual or cognitive impairments throughout the study. Compared to the first time the sharks were examined in the Pretraining, the average number of sessions to reach the LC slightly decreased from 12 ± 9 to 12 ± 6 sessions, while the number of sharks reaching the LC (six out of seven) remained the same.

### 3.3. Rays

#### 3.3.1. Group 3

As described before, this group of rays was always trained on a new ratio while the surface area was not controlled for. Individuals that reached the LC in the specific training were additionally tested in transfer tests, where the surface area was controlled for. This procedure examined whether the rays were still able to discriminate the previously learned quantity. In the 5 vs. 2 training, four out of five rays reached the LC in 15 ± 11 sessions. During the 3 vs. 6 training four out of five rays reached the LC in 8 ± 3 sessions. In the 4 vs. 7 training four out of five rays reached the LC in 7 ± 3 sessions and during the 7 vs. 2 training, one ray reached the LC in 13 sessions. While three out of four rays were able to the reach the LC in the 5 vs. 7 training in 7 ± 4 sessions, none of the rays was able to reach the LC to discriminate 7 vs. 6 and therefore no transfer tests were implemented. Pooled as a group within the transfer tests the rays chose the higher numerosity significantly more often when tasked to discriminate 6 vs. 3 (*n* = 4, *p* < 0.01), 7 vs. 4 (*n* = 3, *p* < 0.0001) and 7 vs. 2 (*n* = 1, *p* < 0.02) (Figure 7a). No significant choice for the higher quantity was recorded for 5 vs. 2 (*n* = 3, *p* = 0.052) and 7 vs. 5 (*n* = 1, *p* = 0.263) (Figure 7a).

#### 3.3.2. Group 4

Two rays were successfully trained to discriminate 4 vs. 1 and one was successfully trained to discriminate 5 vs. 2 (Table 1). An exemplary graph showing the performance of one ray during the training and while the transfer tests were added to the training scheme can be found in Figure 8. As a group, the rays chose the higher numerosity significantly more often when tasked to discriminate 3 vs. 1 (*n* = 3, *p* < 0.0001), 3 vs. 2 (*n* = 3, *p* < 0.00002), 5 vs. 2 (*n* = 2, *p* < 0.0002), 4 vs. 3 (*n* = 3, *p* < 0.0002), 5 vs. 3 (*n* = 3, *p* < 0.00001), 7 vs. 4 (*n* = 3, *p* < 0.00001), 7 vs. 5 (*n* = 3, *p* < 0.003), 9 vs. 5 (*n* = 1, *p* < 0.05), 12 vs. 6 (*n* = 2, *p* < 0.0001), 15 vs. 5 (*n* = 2, *p* < 0.001) and 12 vs. 8 (*n* = 3, *p* < 0.04) (Figure 7b). No significant choice towards the higher numerosity was recorded for 5 vs. 4 (*n* = 2, *p* = 1), 7 vs. 6 (*n* = 1, *p* = 1) and 12 vs. 9 (*n* = 2, *p* = 0.6358) (Figure 7b). The 7 vs. 4 and 9 vs. 5 transfer test were additionally used to see whether the rays used a “relative” (higher or lower quantity) or “absolute” (“4” or “5” depending on what they were trained on) approach to discriminate quantities. In both tests, the rays choose the higher numerosity (“relative strategy”) significantly more often than the absolute number of items they were trained on. Furthermore, they were tested whether the size of the stimuli influenced their choice. Therefore, the negative stimuli were downscaled, with individual symbol size being smaller than the individual symbol size of the positive stimuli. The rays still showed a significant choice towards the higher quantity on which they have been trained on (4 vs. 1 or 5 vs. 2) (*n* = 3, *p* < 0.000001).

## 4. Discussion

The present study examined the ability of *C. griseum* and *P. motoro* to discriminate two different quantities of two-dimensional objects. The results of the experiments show that individuals of both species successfully solved the training tasks based on numerical information while continuous variables were controlled for. The results corroborate findings from several other publications examining this ability in fish over the last decade [35,60]. Previously, the ability to discriminate quantities had only been examined in Port Jackson sharks (*Heterodontus portusjacksoni*) [56]. Unfortunately, continuous variables were not controlled for. Nonetheless, similar to the present study it was shown that there was a high degree of individual variability in regard to learning performance and the development of side preferences in those sharks, that did not successfully perform during training. In the current study, this resulted in only a small number of individuals (7 out of 15 sharks, 7 out of 14 rays) reaching the stage where transfer tests were implemented. Especially shark group 2 experienced great difficulty to learn the task. Within all groups, side preferences (either side) were noticeable. In those cases where side preferences persisted and prevented animals from reaching the learning criterion, individuals were assumed to not have comprehended the task. However, having a side preference can generally be considered a reliable replacement strategy, as it enables an animal to secure 50% of the rewards. The low number of individuals advancing to the transfer test stage may not only reflect individual learning variation but could also indicated that numerical discrimination abilities may not be as fundamental to these species as to others [7,13]. Information on the ecology and the natural behavior of *C. griseum* and *P. motoro* is scarce and whether there is a need to discriminate quantities for mating, foraging or other behaviors is not known [61,62].

The following discussion is divided into sharks and rays. As shark group 2 was not examined in transfer tests at all, the variety of transfer tests for *C. griseum* differed compared to the ones used for *P. motoro*.

### 4.1. Grey Bamboo Shark (Chiloscyllium griseum)

As expected, the two shark groups performed successfully in the Pretraining test, having shown consistenly high performances in previous cognition studies and in various setups [52,59]. However, no individual of shark group 2 reached the LC within their discrimination training sessions, and therefore no transfer tests were conducted. The following discussion is therefore based on the performance of group 1. Seven out of eight individuals were successfully trained on a 4 vs. 1 task. In the following transfer test, the effect of the only potentially confounding and uncontrolled variable during training was tested; i.e., thesize of the single negative stimulus was adjusted to the size of a single positive stimulus. All sharks still managed to choose the higher numerosity significantly more often. This shows that the sharks actively chose the higher numerosity during training instead of basing their choice on stimulus size. However, the low numerosity and “easier” ratio of 0.25 (4 vs. 1) might have diminished the possible effect of evading a larger (size-wise) negative stimulus. After animals were trained in 4:1, they proceeded to a 5:2 training. Here it showed that more than half of the sharks had problems reaching the LC. This result matches the observations by Vila Pouca et al. (2019) and underlines high individual differences [56]. Only three sharks managed to successfully reach the LC in the 5:2 task and maintained a high performance, so that transfer tests were conducted. First, a set of transfer tests was applied to evaluate the possible influence of shading and individual stimulus size, and the ability to transfer training skills to new shapes. Removing the coloration of the stimuli excluded the sharks’ option to make a choice based on the higher numerosity of shading gradients within the stimulus set. Additionally, the cumulative surface area of the positive stimulus was reduced to half of the negative one, and in the last transfer test, unfamiliar non-geometrical stimuli were presented. All sharks successfully chose the higher quantity in all transfer tests. This could be a seen as first evidence that *C. griseum* can rely on numerical information and is able to transfer and apply the previously learned information onto an alteredtask.

The subsequent transfer tests only included a change in numerosity and ratio, the three sharks were tested on their ability to discriminate 5 vs. 3, 5 vs. 4, and 7 vs. 4. All ratios fall within the range of the hypothesized border area between the OFS and ANS system [15,20]. Sharks were able to successfully discriminate 5 vs. 3 and 7 vs. 4 but not 5 vs. 4. The results match the findings of other studies in fish, mammals, and amphibians [35,63,64]. OFS limits have shown to be at approximately 3 for the angelfish, goldbelly topminnow and redtail splitfin [65,66,67], 4 in primates (*Macaca mulatta*) and possibly 5 in guppies [1,68,69]. The present data could indicate that the OFS in *C. griseum* is limited to numbers of 4 or below. The 7 vs. 4 task is likely within the range of the ANS. Future studies could build upon the data from this study and investigate the limit of the two systems in this species further. The combined results show that *C. griseum* is able to use numerical information when solving quantity discrimination tasks, and its performance is comparable to those of other species. Whether numerical abilities are advantageous for *C. griseum* in the wild cannot be answered from artificial laboratory experiments and too little information is available about its natural behavior to speculate on this. In light of the astounding results of other cognition studies on the grey bamboo shark (e.g., [41,47,51,53]) and considering how few animals eventually proceeded to the transfer phase, it can be argued that, this ability is unlikely to be crucially important to the species.

### 4.2. Ocellate River Stingray (Potamotrygon motoro)

Just like the sharks, both ray groups successfully underwent Pretraining. Group 3 was subsequently trained in a series of consecutive discrimination tasks and then, following each training, tested in matching transfer tests. In contrast, the individuals of group 4 were only trained once (4 vs. 1 or 5 vs. 2) and then examined in many transfer tests using a wide range of different ratios. The transfer tests implemented for both groups varied to maximize the number of different ratios and quantities tested for this species. Individuals of group 3 were successfully trained to discriminate 6 vs. 3, 7 vs. 4, 7 vs. 2, 5 vs. 2, and 7 vs. 5 during the consecutive training procedures. None of the rays was able to discriminate 7 vs. 6. In the unrewarded transfer tests, rays discriminated 6 vs. 3, 7 vs. 4, and 7 vs. 2. This shows that this group of *P. motoro* was able to discriminate quantities up to a ratio of 0.57. Surprisingly, the rays failed to discriminate 5 vs. 2 (0.4) and expectedly failed to discriminate 7 vs. 5 (0.71). Nevertheless, while not significant, the three individuals tested for 5 vs. 2 showed a tendency towards choosing the correct quantity (*p* = 0.052). The findings correspond to those found for shark group 1 and the study by Villa Pouca et al. [56]. The results of group 4 are based on three individuals that were continuously trained in a 4 vs. 1 (*n* = 2) and 5 vs. 2 task (*n* = 1). Transfer tests were implemented during overtraining, i.e., after the LC was reached. Due to their high and reliable performance (Figure 7), 16 different transfer tests were conducted. While all continuous variables were controlled for, the rays managed to discriminate quantities up to a ratio of 0.75 in a lower numerosity (4 vs. 3) and 0.66 in a higher numerosity (12 vs. 8). Comparable studies in fish showed a similar ability when distinguishing higher and lower numerosities [39,40]. *P. motoro* has outperformed the results reported in infants [70] and angelfish [39] while showing similar performance compared to primates [1]. Only guppies were able to discriminate 4 vs. 5 after extensive training, a task the stingrays failed to achieve in the transfer test [69]. Another question, that transfer test results of this group answered, was whether rays used an ‘absolute’ or a ‘relative’ strategy [37]. It was examined whether the rays chose the higher numerosity based on choosing the “absolute” number ‘4’ (4 vs. 1 training) or ‘5’ (5 vs. 2 training) or if they simply choose the larger one of two quantities (relative). Transfer tests were conducted were the two strategies were placed into conflict, i.e., the higher training numbers were now the lower numbers of the two transfer test alternatives (7 vs. 4 and 9 vs. 5). All three stingrays used a “relative strategy” to distinguish between the different ratios provided.

When analyzing the results of both stingray groups, individual differences seem to be playing an important role, just like in the shark groups. While individuals of group 3 could not discriminate 5 vs. 2 and 7 vs. 5, the three rays of group 4 could. Individual differences are present in many species and need to be accounted for when analyzing and comparing data [71,72]. Even the design of the experimental setup has shown to have an influence on numerical abilities in fish like goldbelly topminnows (*Girardinus falcatus*) and guppies (*Poecilia reticulata*) [66,73]. Within group 4, *P. motoro* was able to discriminate 3 vs. 1 and 3 vs. 2, that fall within the range of the OFS [15,20]. While the stingrays as well as the sharks, were able to discriminate 5 vs. 3 and 7 vs. 4, the rays failed to do so for 5 vs. 4 and 7 vs. 6. Again, this seems to indicate the transition from the OFS to the ANS, in which a difference of one item is not sufficient for distinction anymore. The range of the OFS found for the stingrays therefore appears to be one to four, which is similar to other animals tested [35,63,64]. Within the ANS range, the rays managed successfully to discriminate 9 vs. 5, 12 vs. 6 and 12 vs. 5 in addition to the previously mentioned 12 vs. 8 task. Increasing the ratio to 0.75 in a 12 vs. 9 task, resulted in none of the individuals performing successfully. This matches previous findings by Agrillo et al., where guppies, zebrafish, redtail splitfin, and Siamese fighting fish could discriminate 12 vs. 8 but not 12 vs. 9 [39]. The limits of the ANS and the range of the OFS for sharks should be investigated further, and additional studies on other shark and ray species may show how numercal abilities are generally distributed within the Chondrichthyes.

## 5. Conclusions

The data gathered within this study shows that the ability to discriminate quantities of two-dimensional objects based on numerical information alone is present not only in teleosts but also in elasmobranchs. Intraspecific differences and a considerably high number of unsuccessful individuals indicate though, that despite being able to ‘count’ (by pick the larger of two quantities within a particular range) numerical abilities may not be of particular importance to both species. However, individuals that passed the training stages maintained very high-performance levels and successfully showed impressive abilities to extrapolate learned knowledge to new tasks. The OFS is likely to range between 1 and 4 and both species preferred using a relative (choosing the larger quantity) over an absolute (e.g., always choosing ‘5’) strategy. Results confirm previous findings that elasmobranchs possess many of the same cognitive abilities and to a similar extent as other vertebrates.

## Figures and Tables

**Figure 1 animals-11-02634-f001:**
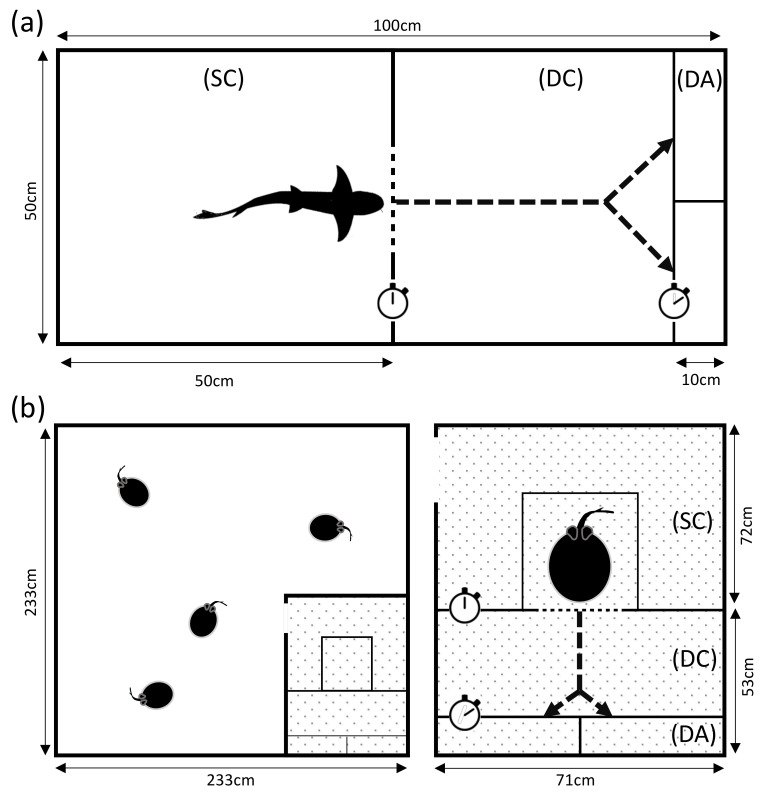
Overview of experimental tanks for rays and sharks. Both experimental tanks are divided into a “Starting Compartment” (SC), “Decision Compartment” (DC) and a “Decision Area” (DA). The dotted line between the SC and DC indicates the hand-operated guillotine door. Stimuli were presented on the shorter side of the experimental tank within the DA. The dotted arrows mark a simplified trial; (**a**) separate experimental tank of the sharks. (**b**) Home tank of the stingrays where the experimental tank (dotted area) was integrated which also holds a starting box for the animals within the SC.

**Figure 2 animals-11-02634-f002:**
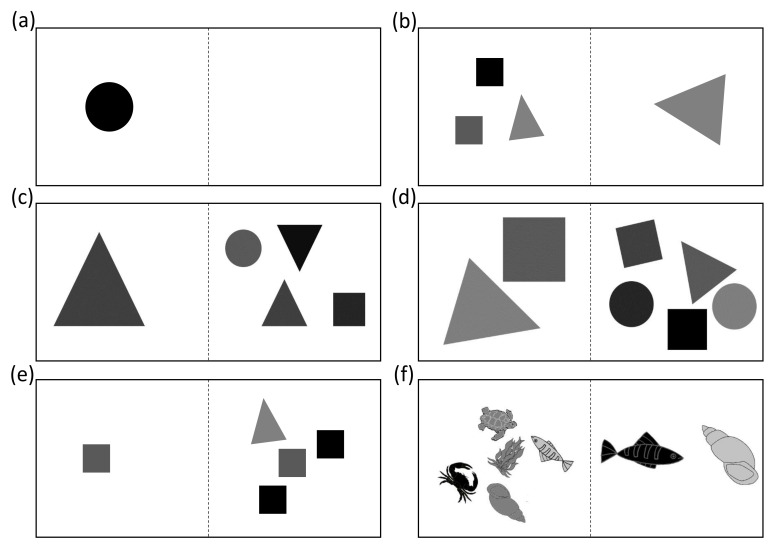
Example slides for the numerical discrimination training and transfer tests. (**a**) Pretraining (**b**) 3 vs. 1 Training (**c**) 4 vs. 1 Training (**d**) 5 vs. 2 Training (**e**) Transfer test with same-sized stimuli (**f**) Transfer test with non-geometrical stimuli. Except for the Pretraining, all slides shown are just representative cards (out of at least 40 for each pair) and were controlled for continuous variables. The dotted line indicates the area in which a divider separated the left from the right side and was not present on the actual stimuli pairs.

**Figure 3 animals-11-02634-f003:**
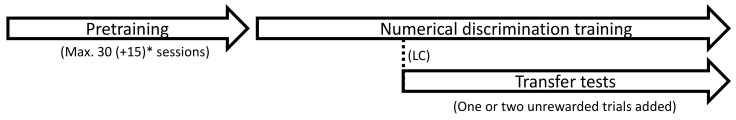
Schematic overview of the experimental timeline. After the successful “Pretraining” animals were transferred to a specific “Numerical discrimination training”. When reaching the LC within the numerical discrimination training, unrewarded transfer tests were implemented and added to the training. * Experimentally naïve individuals that did not reach the LC before the end of the 30th session, were given an additional 15 sessions after which their performance was analyzed.

**Figure 4 animals-11-02634-f004:**
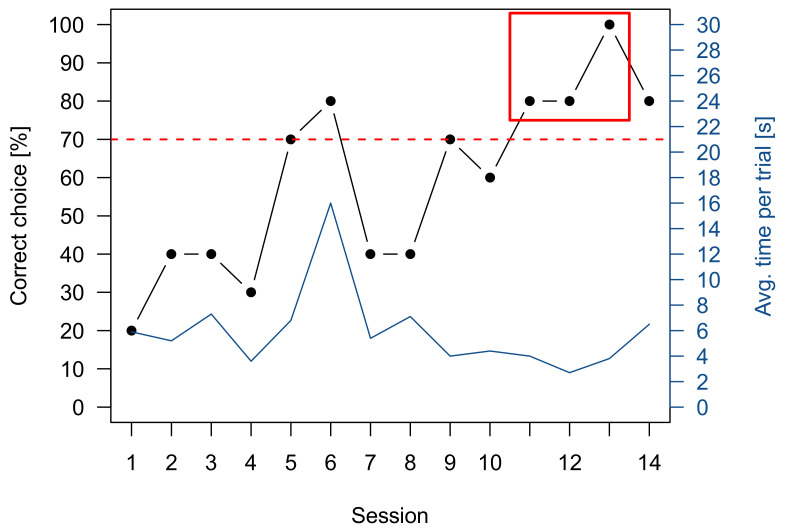
Exemplary learning curve of a shark during Pretraining. The black dots show the percentage of correct choices per ten-trial session. The red dotted line indicates the lower limit of the LC while the red square highlights the section in which the LC was achieved. The blue line gives the average trial time for each session.

**Figure 5 animals-11-02634-f005:**
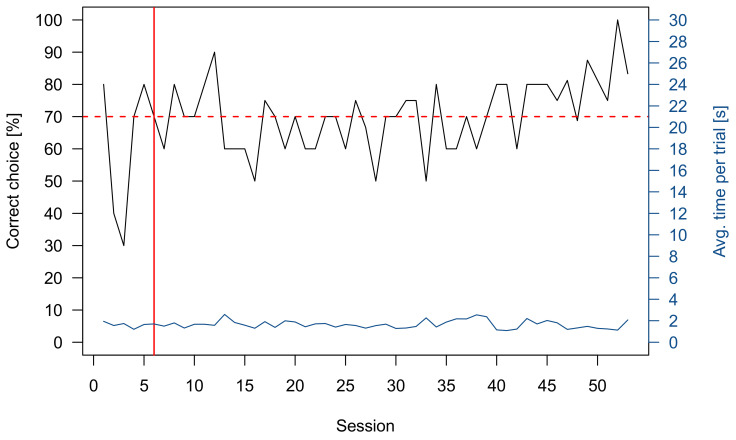
Exemplary graph of the performance of a shark during the entire training and testing phase. The black line shows the percentage of correct choices per ten-trial session. The horizontal red dotted line indicates the lower limit of the LC, while the vertical red line indicates when the LC was achieved. In the sessions following the red vertical line, transfer tests were added to the regular ten trial session. The blue line gives the average trial time for each session.

**Figure 6 animals-11-02634-f006:**
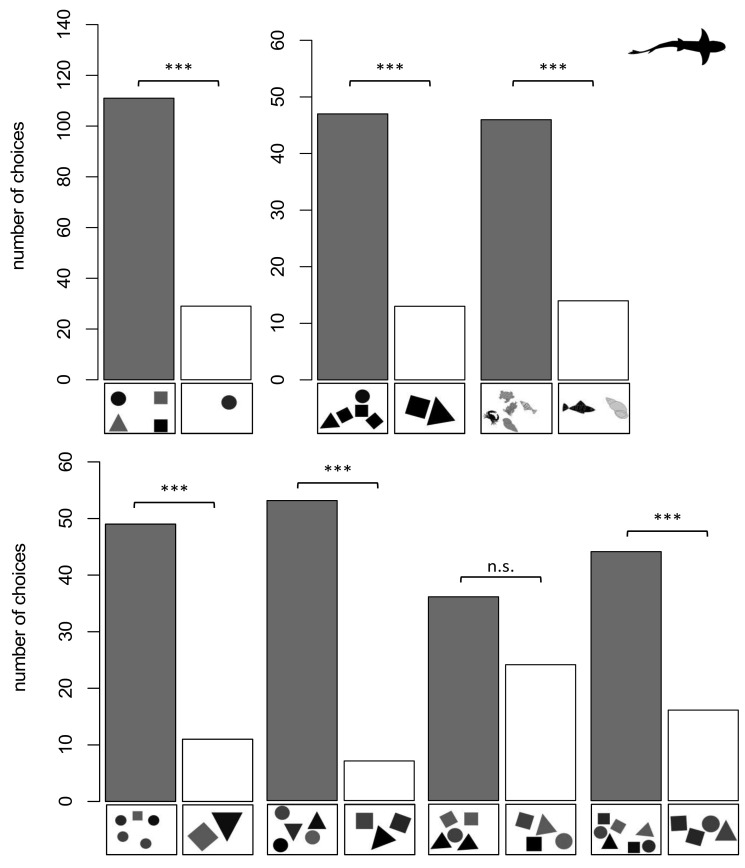
Overview of shark group 1 results in the transfer tests. On the X-Axis exemplary stimulus-pairs are displayed. The dark bars represent the correct choice for higher numerosity. *p* > 0.05 not significant (n.s.), *p* < 0.001 significant (***).

**Figure 7 animals-11-02634-f007:**
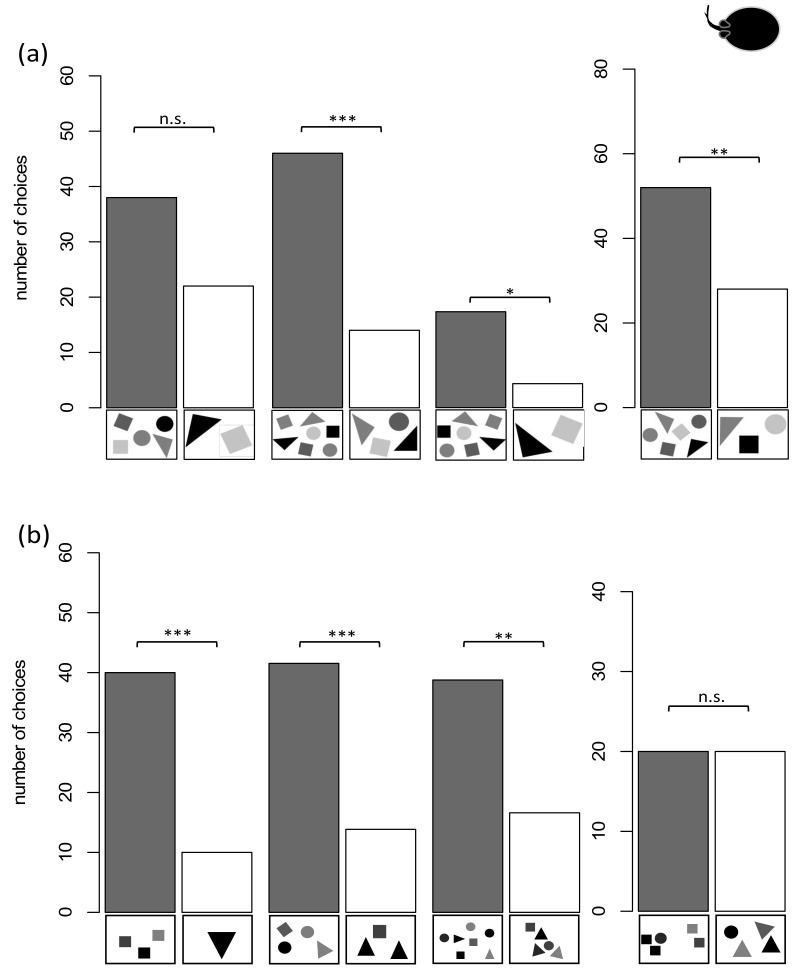
Overview of stingray groups 3 and 4 and their results in the transfer tests. On the X- Axis exemplary stimuli-pairs are displayed. The dark bars represent the correct choice for higher numerosity. *p* > 0.05 not significant (n.s.), *p* < 0.05 significant (*), *p* < 0.01 significant (**), *p* < 0.001 significant (***). (**a**) Results for group 3, (**b**) Results for group 4.

**Figure 8 animals-11-02634-f008:**
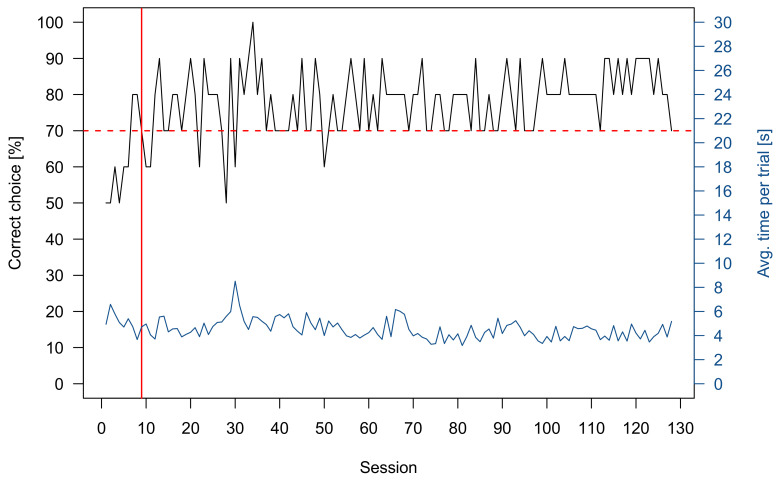
Exemplary graph of the performance of a ray during during the entire training and testing phase. The black line shows the percentage of correct choices per ten-trial session. The horizontal red dotted line indicates the lower limit of the LC, while the vertical red line indicates when the LC was achieved. In the sessions following the red vertical line, transfer tests were added to the regular ten trial session. The blue line gives the average trial time for each session.

**Table 1 animals-11-02634-t001:** Overview of both species performance in the transfer tests. * Two individuals of Group 4 were able to discriminate 5 vs. 2 while three individuals of Group 3 were not. ** Three individuals of Group 4 were able to discriminate 7 vs. 5 while one individual of Group 3 was not.

	*C. griseum*	*P. motoro*
Groups tested within the Transfer tests	Group 1	Groups 3 and 4
Geometrical two-dimensional objects presented as stimuli	Circle, Square, Triangle	
Tasks within the Training procedures	1 vs. 4 2 vs. 5	1 vs. 3 (Group 4)
		1 vs. 4 (Groups 3 and 4)
		2 vs. 5 (Groups 3 and 4)
		3 vs. 6 (Group 3)
		4 vs. 7 (Group 3)
		5 vs. 7 (Group 3)
		6 vs. 7 (Group 3)
Successfully discriminated quantities in Transfer tests with controlled continuous variables	4 vs. 1 5 vs. 2 (Same stimuluscolor) 5 vs. 2 (Non-geometrical stimuli) 5 vs. 2 (Positive stimuli reduced in size) 5 vs. 3 7 vs. 4	3 vs. 1
		3 vs. 2
		4 vs. 3
		5 vs. 2 (Group 4) *
		5 vs. 3
		6 vs. 3
		7 vs. 2
		7 vs. 4
		7 vs. 5 (Group 4) **
		9 vs. 5
		12 vs. 6
		12 vs. 8
		15 vs. 5
Unable to discriminate quantities in Transfer tests with controlled continuous variables	5 vs. 4	5 vs. 2 (Group 3) *
		5 vs. 4
		7 vs. 5 (Group 3) **
		7 vs. 6
		12 vs. 9
Use of “relative” or “absolute” approach to discriminate quantities	Not tested	“Relative strategy”

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
