# Peer review of "Counting on Numbers—Numerical Abilities in Grey Bamboo Sharks and Ocellate River Stingrays"

_animals, 2021, doi:10.3390/ani11092634_

Round 1
Reviewer 1 Report
(See attached review report)

Reviewer 2 Report
The present study “Counting on numbers – Numerical abilities in grey bamboo sharks and ocellate river stingrays by Kreuter and colleagues (ID animals-1305588)” provide evidence of numerical skills in two species of elasmobranchs, the first evidence in a non-teleosts species. Both species, the grey bamboo shark (Chiloscyllium griseum) and the ocellate river stingray (Potamotrygon motoro), were demonstrated to discriminated between controlled quantity to transfer the majority of the non-numerical information, i.e. continuous variable, then they solve new tasks accordingly to the learned numerical rule. Indeed, the authors provided a comparative review of numerical capacities among fish species. The idea to sum up current results in this field seems appropriate.
I am sincerely appreciated the intelligibility of this study. I consider the present study as suitable for a publication. My main concern is that some fundamental literature on numerical cognition are not mentioned, and it needs more attention. Moreover, authors should correct errors thorough the text.
My main issues concern the authors lack to mention the hypothesis of a single system, which is involved in processing both small and large numerosity (see Gallistel and Gelman 2000; Brannon and Roitman 2003; Cantlon et al. 2009). Authors mentioned the “two system hypotheses”, concerning the existence of two distinct system which are activated when a specific object are presented. Briefly, the object file system (OFS) is designated to process small range numerosity (e.g., range 1–4 items), while the approximate number system is activated when presenting large numerosity ranges (e.g., > 4 items). Despite the wide inter- and intra-species variation of numerical competence as authors correctly reviewed (see also Agrillo et al., 2015), many researchers are more inclined to accept the existence of a single core system for processing quantitative information (e.g., Beran et al. 2015; Cantlon and Bran 2007; Rugani et al. 2013). Authors should mention both hypothesis in order to provide a full completed scene of the current literature.
Minor comment
Line 7: “the cognitive ability to discriminate quantities” can be summaries as “quantitative discrimination abilities”
Line 13-15: the sentence seems incompletely. What is the contribution of your study to the present knowledge?
Line 26: remove the comma. In addition, I would not use “relevant”. It has been provided that several species mainly used a mixture of information to discriminate between quantities (e.g., continuous and discrete quantities; Davis and Pérusse 1988). Indeed, the high degree of individual variations may be attributed to the methodologies adopted in this study.
Line 34: “that” should be “and”
Line 53-63: I would recommend creating a new paragraph in which authors provided a short description of the difference between continuous and discrete quantities.
Line 62: species names are missing
Line 69: “basic ability” is not appropriate. I would suggest “low cognitive load”.
Line 70: the two species investigated in the current study have been previously in the abstract and not in the main text.
Line 94-95: species name should be in italics
Line 102: this section should be separated. In addition, authors should consider reporting a table in which methodological information (typology of task, stimuli etc.) are summarized for each group.
Line 187-190: if authors used a one-tailed test, they would consider only a specific direction for the effect and not either the direction (i.e., negative or positive stimulus).
Line 207: “understanding” should be substituted with “habituation”
Reference
Agrillo C, Miletto Petrazzini ME, Bisazza A (2015b) At the root of math: numerical abilities in fish. In: Geary DC, Berch DB, Koepke KM (eds) Mathematical Cognition and Learning Vol. 1. Elsevier, Amsterdam, NL, pp. 3-33. https://doi.org/10.1016/B978-0-12-420133-0.00001-6
Beran MJ, Parrish AE, Evans TA (2015) Numerical cognition and quantitative abilities in nonhuman primates. In: Geary DC, Berch DB, Koepke KM (eds) Mathematical Cognition and Learning Vol. 1. Elsevier, Amsterdam, NL, pp. 91-119. https://doi.org/10.1016/B978-0-12-420133-0.00004-1Ge
Brannon EM, Roitman JD (2003) Nonverbal representations of time and number in animals and human infants. In: Meck WH Meck (ed) Functional and neural mechanisms of interval timing. CRC Press Routledge Taylor & Francis Group, Abingdon, UK, pp 143-182. https://doi.org/10.1201/9780203009574.ch6
Cantlon JF, Brannon EM (2007) How much does number matter to a monkey (Macaca mulatta)?. J Exp Psychol Anim B 33:32. https://doi.org/10.1037/0097-7403.33.1.32
Cantlon JF, Platt ML, Brannon EM (2009) Beyond the number domain. Trends Cogn Sci 13:83-91. https://doi.org/10.1016/j.tics.2008.11.007
Davis H, Pérusse R (1988). Numerical competence in animals: Definitional issues, current evidence, and a new research agenda. Behav Brain Sci 11:561-579. https://doi.org/10.1017/S0140525X00053437
Gallistel CR, Gelman R (2000) Non-verbal numerical cognition: From reals to integers. Trends Cogn Sci 4:59-65. https://doi.org/10.1016/S1364-6613(99)01424-2
Rugani R, Cavazzana A, Vallortigara G, Regolin L (2013) One, two, three, four, or is there something more? Numerical discrimination in day-old domestic chicks. Anim Cogn 16:557–564 (2013). https://doi.org/10.1007/s10071-012-0593-8
